
# Dynamic variability examination of Mediterranean frontogenesis: teleconnection of fronts and flood 2010

**B. A. Munir[1], H. A. Imran[2], and I. Ashraf[3]**

[1]National Weather Forecasting Unit, Aviation Division, Pakistan Meteorological Department, Islamabad, Pakistan
[2]Environmental Protection and Agriculture Food Production, University of Hohenheim, Stuttgart, Germany
[3]Institute of GIS, School of Civil and Environmental Engineering, National University of Sciences and Technology, Islamabad, Pakistan

Received: 18 October 2015 – Accepted: 5 December 2015 – Published: 15 January 2016

Correspondence to: B. A. Munir (bilalahmedmunir35@gmail.com)

Published by Copernicus Publications on behalf of the European Geosciences Union.

**NHESSD**

doi:10.5194/nhess-2015-290

Dynamic variability examination of Mediterranean frontogenesis

B. A. Munir et al.



**NHESSD**

doi:10.5194/nhess-2015-290

Dynamic variability examination of Mediterranean frontogenesis

B. A. Munir et al.

**Abstract**

An improved scheme for the detection of Mediterranean frontal activities is proposed, based on the identification of cloud pattern, thermal gradient and water content of air masses using Meteosat-7 satellite imagery. Owing to highly variable nature of fronts, spatial shift occurring over 1.5 years are analyzed. Full disc imagery of Meteosat-7 satellite is used for the analysis over vast geographical distribution of Eurasia. The study examined the fundamental characteristics of fronts and effects associated with Mediterranean fronts with the analysis for the flood event of 2010 in Pakistan. The results show seasonal as well as annual change in the range of lower and upper latitudinal limits of frontogenesis. Seasons of winter 2011 shows an increase in speed whereas a decrease is observed in summer. The identification process has shown a frontal span over northern areas of Pakistan during flood event of 2010 accentuating the monsoonal rainfall intensity all over the country. The result of this analysis can be used to estimate the behavior of the mid-latitudes global circulations. The anticipated outcome of this research study is the identification of abrupt nature of frontal processes.

# 1 Introduction

The concept of frontogenesis remains the mainstay of meteorological fundamentals at synoptic scale with the introduction in early twentieth century (Bjerknes et al., 1922). Now days, globally the concept of fronts and its understanding remained a succinct method in probable prediction and summarization of weather conditions. Though the term frontogenesis (process of front formation) solely defines the interactive boundary between two air masses with different thermal characteristics, these boundaries are more often associated with significant weather conditions as precipitation etc. This process defines the local conditions of weather, by its changing behavior is with geographical identification on maps or satellite imagery (Berry et al., 2011).

**NHESSD**

doi:10.5194/nhess-2015-290

**Dynamic variability examination of Mediterranean frontogenesis**

B. A. Munir et al.

Generally, fronts are characterized by intense thermal gradients and considered as highly dynamic zones persistently changing through both space and time. The dynamic nature of these features is due to the adjacent air masses with conflicting natures. Apart from the atmospheric factors, topographical features also show influence on weather phenomenon including coastal cyclogenesis (Cione et al., 1993), Piedmont fronts (Businger et al., 1991), coastal frontogenesis (Riordan, 1990) and weak, non-classic frontal boundaries (Vescio et al., 1993) depending upon the seasonality of the year.

Climatologically oceanic frontogeneses are considered important as these phenomenons plays a substantial role in global ocean atmosphere interactions. According to O'Neill et al. (2003) and Chelton et al. (2004) fronts exhibit nonlinear flows and processes on a range of different temporal and spatial scales, hence are variable in strength, intensity and other meteorological characteristics. Due to the dynamic nature of these features successful detection and monitoring therefore is a nontrivial problem. Geographically, subtropical frontal activities and upper tropospheric cyclonic vortices of Northern Hemisphere have received considerable attention (Palmer, 1951; Simpson, 1952; Riehl, 1954, 1977; Ramage, 1962). Whereas, large-scale oceanic frontal zones (OFZs) remains prominent features maintained by the heat fluxes distribution. These processes are of high variable nature with the importance for understanding the ocean land thermodynamics that remained under several investigations since mid-1970s (Hopkins et al., 2008; Nieto et al., 2012).

Weather predictions using satellite imagery revolute with the launch of the first weather satellites in early 1960s. Traditions were set with the advent of interpretation tools on individual albedo values and brightness temperature measurements on pixel level. However, meteorological experts are able to interpret the imagery in a vast exposure of weather information through variable fundamental parameters including; shape, size and texture of various clouds and cloud systems, and also by exploring their context in relation to topography and other weather systems (Pankiewicz, 1995).

Weather forecasting by exploring the features of satellite imagery has great importance that occurs on a variety of scales ranging from local to global. Modern satellite technologies particularly images from remote sensing techniques provide opportunities to detect a full range of patterns, from their differences in surface reflectance or their intrinsic properties. With the invention of high accuracy satellite product and processing techniques, it would seem possible to classify the cloud type, cloud pattern (vortices, lee waves and comas), and meteorological objects and processes such as trough, frontal bands and cyclogenesis, frontogenesis respectively in a satellite image with a setup of automatic systems. However thermal fronts are more often studied weather phenomena due to the universal nature of fronts in both open and coastal areas of any ocean as they are associated to a wide variety of circulation patterns (Larry et al., 2007; Pankiewicz, 1995).

The following study focused over the life cycle of fronts and its behavior that is not consistent with time. All efforts made to explore the hidden monopoly of the nature of the processes of frontogenesis.

It is surprising that despite common use, to our knowledge, no comprehensive climatology of fronts currently exists in the literature. The Mediterranean frontogenesis shows a direct impact on the European and Asian weather. These climatic phenomena of frontogenesis result in heavy precipitation in underlying areas. To access the variable nature of fronts, the study seeks to address the following objectives in spatial context.

1. Spatio-temporal variability assessment of Mediterranean frontogenesis

2. Make use of slope and speed factor to assess the intensity of frontogenesis

3. Seasonal analysis of rainfall for the spatio-temporal limits of frontogenesis

## 2   Material and methods

The study area for the examination of the variable nature of frontal activities focuses on the eastern longitudinal limits of Mediterranean Sea (source). The main focus was

**NHESSD**

doi:10.5194/nhess-2015-290

**Dynamic variability examination of Mediterranean frontogenesis**

B. A. Munir et al.

**NHESSD**

doi:10.5194/nhess-2015-290

**Dynamic variability examination of Mediterranean frontogenesis**

B. A. Munir et al.

on the mid-latitudes of the Northern Hemisphere, latitudinal range is (30–65)° N and longitudinal range is (10–100)° E. The limits of the study area were selected as to minimize the distortion of spherical shape of satellites imagery. The study area includes Eurasia including countries; Cyprus, Egypt, Lebanon, Jordan, Syria, Turkey, Israel, Saudi Arabia, Iraq, Armenia, Georgia, Russia, Azerbaijan, Kazakhstan, Turkmenistan, Uzbekistan, Iran, Afghanistan and Pakistan.

Meteosat-7 satellite series secondary product was used for the identification of frontogenesis due to its free availability and for the coverage of full disc image in visible (0.55–1.05 μm), infra-red (10.2–12.5 μm) and water vapor (6.2–7.6 μm) band combination. VISSR (Visible and Infrared Spin Scan Radiometer) is a scanner carried aboard on the GMS (Geo-stationary Meteorological Satellite), stationed over Japan. The GMS imagery is re-broadcasted by Meteosat satellite. Whereas, precipitation data obtained from the derived product of TRMM (Tropical Rainfall Monitoring Mission) was used to analyze the resulting rainfall trends. The precipitation product was used to correlate the effects of frontogenesis in the major areas of interest. Main focus is on rain fall that is the major effect of frontal process.

## 2.1 Image classification

With the designed objectives full disc visualization of globe was needed for which geostationary satellite image is essential. Huge directory of individual band imagery (1.5 year) containing 1635 images were collected from December 2009 to August 2011 with temporal resolution of 1 day. The images were downloaded from the official website of the Dundee Satellite Station (Fig. 1). The images are of the sensor VISSR and the satellite has a slot of 57° E. The individual band was necessary as for the identification of pattern, moisture and thermal gradient present in the clouds.

The unsupervised classification was performed on each image of infra-red and water vapor bands. The identification process was based on the fundamentals of frontogenesis in 2-dimensional domains (sharp boundary of clouds in visible, thermal gradient in infra-red, and presence of moisture content in water vapor imagery) (Pike,



1999); (Volkert, 1999); (Barry and Chorley, 2010); (Dirks et al., 1988); (Garabato et al., 2001); (Nieto et al., 2012).

Overlay scheme was used in ArcGIS environment for the identification of exact geographic locations where these fundamental parameters of frontogenesis meet in visible, infrared and water vapor bands. The overlay scheme was important as the assertion of frontogenesis should meet the three fundamentals.

## 2.2 Shift analysis

On average fronts have life cycles of 2–5 days. The foregoing research reports such frontal spells because these spells define the nature of frontogenesis well and good. Geographical location of the front was calculated on the starting and the ending day with the help of the two grids (default and manual grids). The shift was calculated as to analyze the geographical extent of the abundance of frontogenesis with the source of Mediterranean Sea fed by Atlantic Ocean. Basic formula of the range is used for the calculation of the latitudinal shifts

Geographical Shift = Difference between the position of first and the last day

To calculate the shift of the formation and trajectory of the fronts time series analysis done with certain time span. Finally, the movements of fronts are verified with the weather data included rain fall data for those area where these fronts hit in the areas of Pakistan and cause heavy rain fall in the year 2009, 2010 and 2011. The movement of frontal process was analyzed on daily basis with the calculation of velocity in $°\,day^{-1}$ by using the following equation:

$$\text{Velocity } (°\,day^{-1}) = \frac{\text{Distance in }°}{\text{Time in days}}$$

## 2.3 Rainfall trends

To detect the rainfall patterns TRMM product was downloaded for the distinct spatial extent and range of the identified frontogenesis spells. The data was available for the

Discussion Paper | Discussion Paper | Discussion Paper | Discussion Paper

**NHESSD**

doi:10.5194/nhess-2015-290

**Dynamic variability examination of Mediterranean frontogenesis**

B. A. Munir et al.

**NHESSD**

doi:10.5194/nhess-2015-290

**Dynamic variability examination of Mediterranean frontogenesis**

B. A. Munir et al.

latitudinal limits of 50° N to 50° S. The rainfall trends were analysed for the accumulated time period of 4 days because front has an average life span of 2–5 days. The rainfall patterns for the summer (July–August) and winter (December–January) season were examined to determine the areas and period of rainfall extremities. In keen rainfall trend for the same period was also observed throughout Pakistan on decadal basis to identify the contribution of these detected fronts in accentuating the rainfall pattern that caused flooding in Pakistan during July–August 2010.

## 3   Results and discussions

### 3.1   Analysis of frontogenesis

Unsupervised classification algorithm with overlay scheme results in a directory of images that possesses the required fundamentals of fronts. The classified schemes shows five classes tiered on the basis of moisture content in the clouds and temperature of the clouds in water vapor and infra-red band respectively. It was observe that the abundance of clouds with moisture content and thermal gradient was mostly found in the tropical regions as these latitudes receive most perpendicular sun rays (Fig. 2) (Barry et al., 2009). The shift towards high latitudes was analyzed in summer due to the expansion of ITCZ. The ITCZ is a convergence field between the opposing trades that shows a seasonal variation just as do the trade winds.

It was scrutinized that different air masses scheme surrounds Eurasia throughout the year including maritime arctic (mA), maritime polar (mP), continental arctic (cA), continental polar (cP), maritime tropic (mT), and continental tropic (cT). The source regions, trajectories, and natural properties of individual air mass discerned the impacts associated with it. High abundance of moisture and thermal gradient was observed in cT during winter and summer with a source over North Africa (Aerographer, 2003).

## 3.2 Geographical extent of frontogenesis

The result shows dynamicity of frontogenesis for years 2010 and 2011. It was observe that geographical distribution and movements of frontal activities in 2010 cover vast area as compare to the year 2011 for the seasons of summer, winter, and spring. The frontal schemes in the mid-latitudes of the Northern Hemisphere, showed a hasty behaviour which was not consistent over time. In the winter 2011 latitudinal range shifts towards higher latitudes, as compare to winter 2010. Upper and lower limits both changes enormously as shown below in Fig. 3. The annual shift in latitude and longitudes also demonstrate the pattern of westerly waves that enforce the movement of frontogenesis in tropical and exra-tropical regions. However, one of the probable reasons is the change in global climate which is keep on with the passage of time resulting in squeezing and harshness in the seasonality (Ahrens, 2011).

The results showed that the process of frontogenesis appears in the latitudinal range of approximately 30–50° in summer 2010 that reduces to approximately 42–58° in summer 2011 with an upward shift of 12° in lower limits. Similarly winter of 2011 the limits shifts toward higher latitude as compare to winter of 2010 (Fig. 4).

The abundance of frontogenesis in winter is most frequent off the coasts of continents in areas of 30–60° latitude. The process could also be instigated over land, but temperature gradient is greater between the ocean and continent, especially in winter, therefore the coastal areas are considered to be more favourable for frontogenesis. However, the frontogenesis is not as abundant and intense in the summer as in the winter because of a declining temperature gradient between the air masses (Aerographer, 2003); (Ahrens, 2007). The changing behaviour of air masses with seasons in terms of intensity and trajectories affects the geographical location and extent of frontal activities. For Example: cT in winter generated from North Africa covers central and western Europe and considered as the major source of heat resulting in cyclonic storms in winter and spring seasons. Whereas cT air mass in summer covers

Discussion Paper | Discussion Paper | Discussion Paper | Discussion Paper | Discussion Paper |

**NHESSD**

doi:10.5194/nhess-2015-290

**Dynamic variability examination of Mediterranean frontogenesis**

B. A. Munir et al.

North Africa, Asia Minor, and South Balkans. This dynamic nature of air masses is cloned and observed for the frontogenesis in mid-latitudes.

In winter 2010 fronts formed in wide longitudinal range of almost 33° (33–66°) easts, while in 2011this range decreases to 12° (55–67°) easts. However, in summers the two years (2010, 2011) show opposite behavior in winter. In 2011 record shift of 49° (20–69°) east was observed in the month of July, while in the summer of 2010 longitudinal shift is 27° (44–71°) east as shown in Fig. 3. The weather condition of a frontal system is well understood by analyzing the speed factor of frontal spell. Fast moving fronts usually cause more severe weather than slow moving fronts. Slow moving fronts, on the other hand, may cause extended periods of unfavorable weather (Aerographer, 2003; Barry et al., 1992). The results showed an inverse pattern of frontal speeds in winter and summer seasons for the respective years of 2010 and 2011. The average frontal speed of 5.2° $day^{-1}$, and 2.5° $day^{-1}$ was observed in winter 2010 and 2011 respectively. However, increased average speed in winter by 0.9° $day^{-1}$ was observed in 2011 as compared to the year 2010 (Fig. 5).

Demonstrating the fundamental characteristics of frontal process, the recorded spells also shows pattern of retrograde motion in a spell of August 2010 in the vicinity of Pakistan. The motion was due to pressure disturbances in the path of the frontal scheme. The spell with a life span of 4 days shows 65, 63, 66, and 71° E longitudinal distributions.

### 3.3 Rainfall analysis

With 0.25° spatial and one day temporal resolution precipitation data from TRMM show best suitability to examine rainfall trends in the observed and defined geographical limits of frontogenesis for summer and winter seasons 2010. It was observed that during winter (December 2009–January 2010) rainfall maxima of 137 mm was experienced in Cyprus during 5–8 December which reached up to 207 mm from 9 to 12 December. Turkey and Iran were found observing up to 111 mm of rainfall from 17 to 20 December. While Cyprus, Lebanon, Syria and turkey were observed,

NHESSD
doi:10.5194/nhess-2015-290

**NHESSD**

doi:10.5194/nhess-2015-290

**Dynamic variability examination of Mediterranean frontogenesis**

B. A. Munir et al.

experiencing maximum rainfall of 134.6 mm during 18–21 January, Cyprus continued experiencing 109 mm of rainfall for extended days from 22 to 25 January. For next 4 days accumulated periods from 26 to 29 January, rainfall maxima trend with 124 mm rainfall spread further North including regions of Turkey, Georgia and Russia. During 6–9 January maximum rainfall of 130 mm was observed in Uzbekistan (Fig. 6).

On contrary, the results shows that during summer season, all maxima of rainfall 130, 160, 225, 190 mm were experienced in Pakistan during 9–12, 17–20, 25–28 July and 2–5 August respectively (Fig. 7).

Two maxima of rainfall during winter season with the average rainfall of 210, 390 mm were experienced during 9–12 July, and 2–5 August respectively in Pakistan. The summer season observed greater number of events in comparison to winter season while the maximum peaks of events with intensive rainfall were found in winter season (Fig. 8).

Pakistan has experienced number of grouped rain events during late July and early August 2010. The rally of rain events results in the worst flooding over past 100 years. Number of studies on Pakistan flood event 2010 (Houze et al., 2011; Webster et al., 2011; Wang et al., 2011) have been emerged (Lau, 2012). The hidden monopoly of disastrous event has been investigated at different aspects including Russian wildfire event and presence of frontogenesis over the north western geographical limits of Pakistan during late July and early August.

According to Lau, 2012 with the usual mundane of global circulations, in summer jet streams being source of movement pushes weather fronts through Eurasia in four to five days. However, in late July and early August 2010, the usual pattern of weather over Russia got stagnated trapping a ridge of high pressure (Fig. 9).

The disturbance in the weather pattern affects flow of jet stream results in slowing down atmospheric system as it moves from east to west and vice versa. The unavoidable changing pattern led the geographical extent of monsoonal activity, high to low pressure, and range of frontogenesis specifically over the territory of Pakistan (Allen, 2011; Lau, 2012). The cold and warm air masses from Siberia and

lower latitudes smashed over Pakistan. The resulted combined system of unusual frontogenesis and monsoonal scheme accentuates the rainfall over Pakistan. The graph of decadal rainfall from 1 July to 29 August in Pakistan indicates the rainfall maxima occurred during second decade of July from 11 to 20, however the maxima shifts towards west in the last and first decade of July and August respectively of Pakistan with the presence of frontal activity (Figs. 10 and 11); Note that this is the same period when Pakistan faced devastating flood scenario. The geographical extent analysis of frontogenesis (see Fig. 3; left) also reveals the presence of frontal spells in late July and early August over North West of Pakistan. The disturbed custom schemes of frontal activities not only aggrandize the temporal range of rainfall season (monsoon) but also magnify the intensity throughout the country.

## 4  Conclusion

The research focuses to analyse the life cycle of the frontal schemes and the rainfall trends associated with the frontogenesis in the mid-latitudes of the Northern Hemisphere. Satellite meteorology with the advancement in remotely sensed techniques have proven the authentic nature for the prediction of weather and identification of meteorological extremes. Based on the data used and techniques involved which results in different outcomes the study concludes the following testimonials:

1. Remotely sensed techniques in satellite meteorology have been proved to be vital and user friendly for the identification, pattern detection, and analyzing the complete trend of frontogenesis.

2. The study verifies the diverse and abrupt nature of the Mediterranean frontogenesis.

3. Global air circulations specifically the jet streams, global temperature, and thermal gradients affect and guide the nature and trend of frontogenesis.

**NHESSD**

doi:10.5194/nhess-2015-290

**Dynamic variability examination of Mediterranean frontogenesis**

B. A. Munir et al.

Discussion Paper | Discussion Paper | Discussion Paper | Discussion Paper

4. Mediterranean Sea, costal and land cover areas show sufficient thermal gradient on average to enhance the frontogenesis in Eurasia between air masses.

5. The study shows that the frontal spells of August (2010) resulted in the accentuation of flood event 2010 in Pakistan. However, rainfall record indicates the presence of Cumulonimbus clouds in the frontal spell that passes over the northern areas of Pakistan in 2010.

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

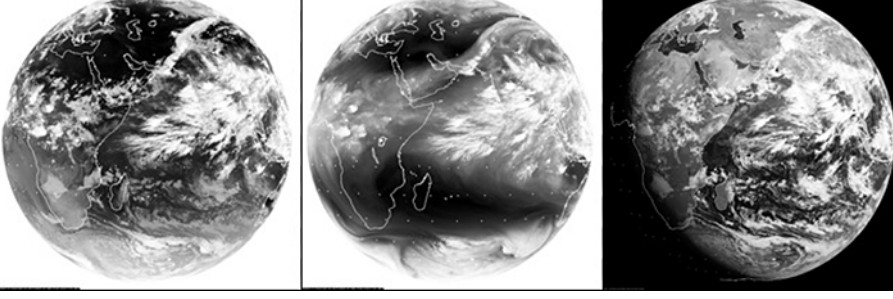

**Figure 1.** Meteosat-7 imagery infra-red (left), water vapor (middle) and visible (right).



**Figure 2.** Classified images of water vapour and infra-red.

**NHESSD**

doi:10.5194/nhess-2015-290

**Dynamic variability examination of Mediterranean frontogenesis**

B. A. Munir et al.

**NHESSD**

doi:10.5194/nhess-2015-290

**Dynamic variability examination of Mediterranean frontogenesis**

B. A. Munir et al.

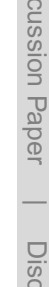

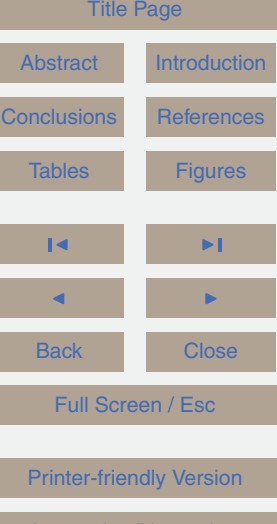

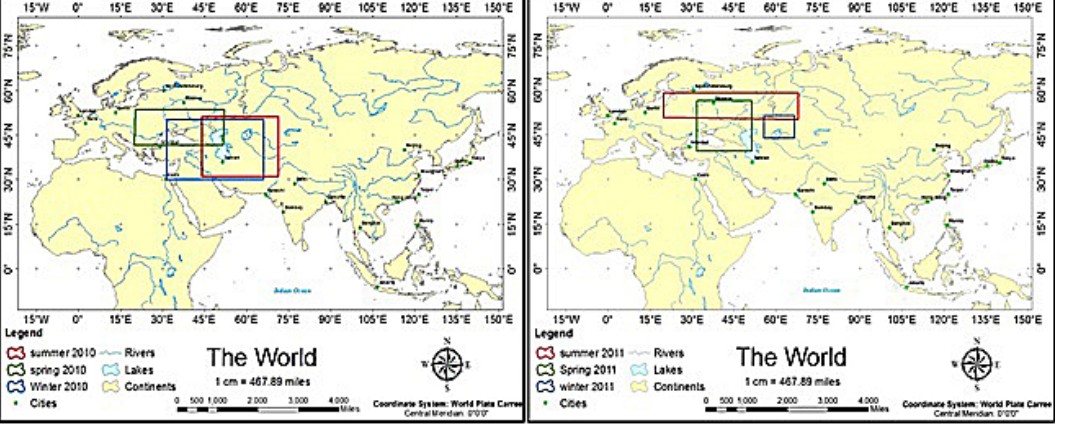

**Figure 3.** Geographical extent of frontogenesis for 2010 and 2011.

**NHESSD**

doi:10.5194/nhess-2015-290

**Dynamic variability examination of Mediterranean frontogenesis**

B. A. Munir et al.

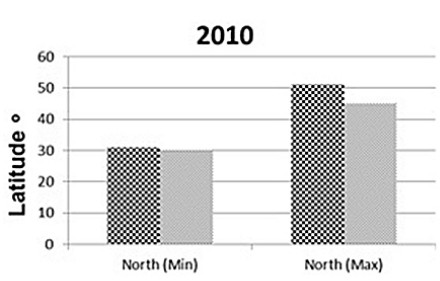
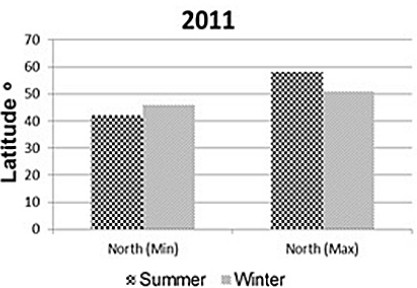

**Figure 4.** Latitudinal limits of frontogenesis for the year 2010 and 2011 in summer and winter.

**NHESSD**

doi:10.5194/nhess-2015-290

**Dynamic variability examination of Mediterranean frontogenesis**

B. A. Munir et al.

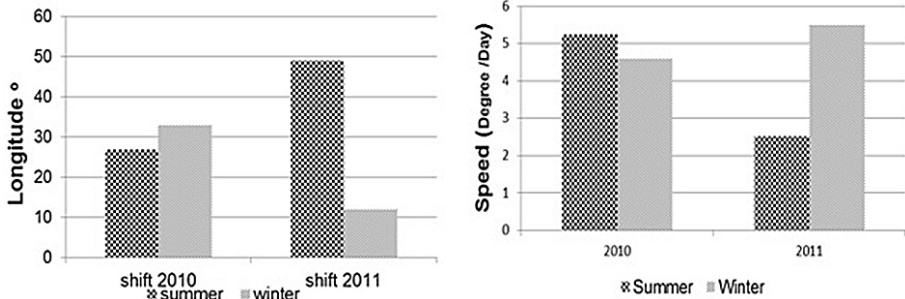

**Figure 5.** Longitudinal shift (left) and speed of front (right) for 2010 and 2011.

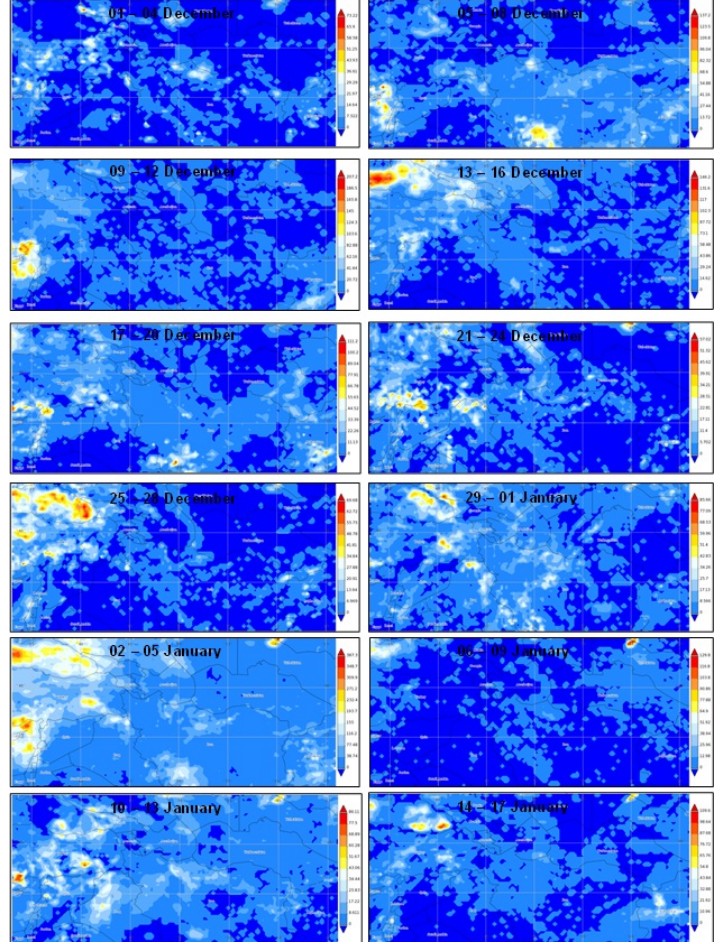

**Figure 6.** Rainfall distribution over geographically delineated extent of frontogenesis in winter 2010.

Discussion Paper | Discussion Paper | Discussion Paper | Discussion Paper

**NHESSD**

doi:10.5194/nhess-2015-290

**Dynamic variability examination of Mediterranean frontogenesis**

B. A. Munir et al.

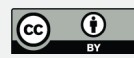

Discussion Paper | Discussion Paper | Discussion Paper | Discussion Paper | Discussion Paper

**NHESSD**

doi:10.5194/nhess-2015-290

**Dynamic variability examination of Mediterranean frontogenesis**

B. A. Munir et al.

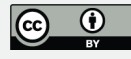

**Figure 7.** Rainfall distribution over geographically delineated extent of frontogenesis in summer 2010.

**NHESSD**

doi:10.5194/nhess-2015-290

**Dynamic variability examination of Mediterranean frontogenesis**

B. A. Munir et al.



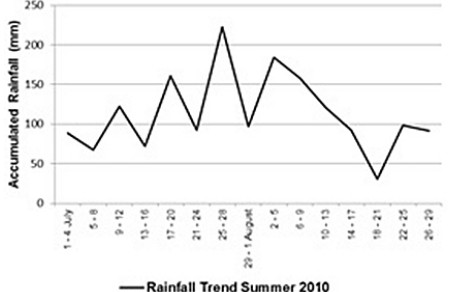 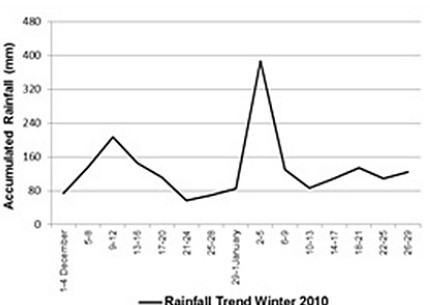

**Figure 8.** Peak four days accumulated rainfall trends for summer and winter 2010.

**NHESSD**

doi:10.5194/nhess-2015-290

**Dynamic variability examination of Mediterranean frontogenesis**

B. A. Munir et al.



**Figure 9.** High and Low distribution over Russia and Pakistan (source: Allen, 2011).

**NHESSD**

doi:10.5194/nhess-2015-290

**Dynamic variability examination of Mediterranean frontogenesis**

B. A. Munir et al.

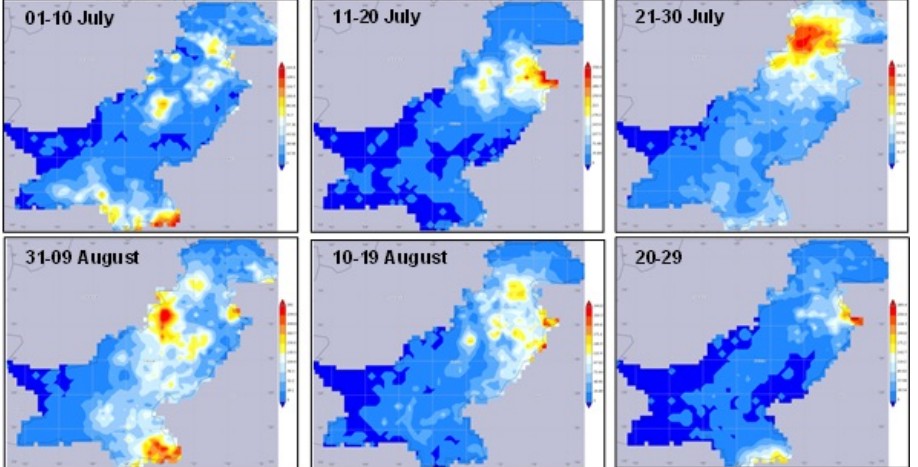

**Figure 10.** Rainfall distribution (mm) over Pakistan during July and August 2010.



**NHESSD**

doi:10.5194/nhess-2015-290

**Dynamic variability examination of Mediterranean frontogenesis**

B. A. Munir et al.

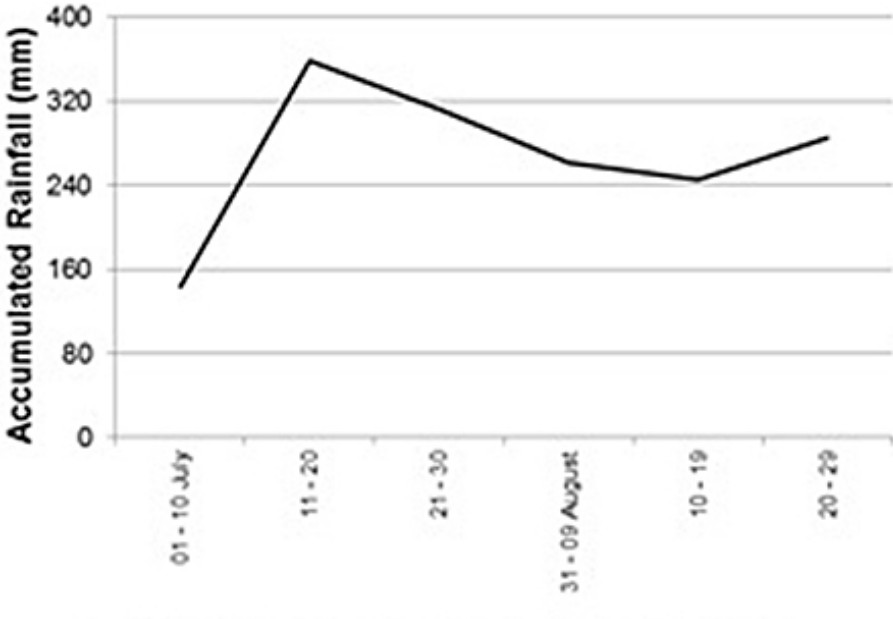

**Figure 11.** Decadal based accumulated peak rainfall for July and August 2010.

