# Peer review of "Dynamic variability examination of Mediterranean frontogenesis: teleconnection of fronts and flood 2010"

_Natural Hazards and Earth System Sciences, 2015_

## Referee Comment (RC1) · Shahid Parvez (Referee) · 21 Feb 2016

Nice research contribution. I endorse the manuscript to be published as it is.

---

## Referee Comment (RC2) · Anonymous Referee #2 · 27 Feb 2016

General comment This paper focuses on the frontal activity in the eastern Mediterranean and western Asia. The study considers the variation of the geographical extent of the fronts for the years 2010-2011 and the relation between the fronts and the rainfall, with a focus on the flooding period on Pakistan occurred in summer 2010.

Specific comments

The main problem with this paper is that the methodology used to identify the fronts is not considered at all. The first sentence of the paper, in the Abstract says: "An improved scheme for the detection of Mediterranean frontal activities is proposed, based on the identification of cloud pattern, thermal gradient and water content of air masses using Meteosat-7 satellite imagery." But this methodology is not presented. The only

sentence about it is: "The unsupervised classification was performed on each image of infra-red and water vapor bands. The identification process was based on the fundamentals of frontogenesis in 2-dimensional domains (sharp boundary of clouds in visible, thermal gradient in infra-red, and presence of moisture content in water vapor imagery)". Which is the improvement, and, more importantly, which is the methodology?

When analysing the pattern of the fronts in 2010-2011 only the findings are discussed but the reader cannot reproduce the results because the methodology used and how it applied is not reported. How the authors conclude that the area selected for 2010-2011 are those where the fronts occurred. Which is the number of fronts detected by the methodology?

The analysis of the correlation of the fronts with the precipitation pattern is based on the qualitative comparison between the TRIMM precipitation patterns with the geographical area covered by the fronts in the years 2010-2011. This correlation is qualitative and more efforts should be done to make it more quantitative.

I suggest a review of the English because there are some errors and few sentences are not understandable.

Minor points See the pdf attached.

Please also note the supplement to this comment:
http://www.nat-hazards-earth-syst-sci-discuss.net/nhess-2015-290/nhess-2015-290-RC2-supplement.pdf
* * *
[Figure]

**Supplement:**

[revised manuscript text omitted]

---

## Referee Comment (RC3) · Anonymous Referee #3 · 7 Mar 2016

In this study, authors use an improved scheme for the detection of Mediterranean frontal activities, based on the identification of cloud pattern, thermal gradient and water content of air masses in Meteosat-7 satellite imagery. The study is conducted over 1.5 years. The flood in 2010 over some areas of Pakistan is analyzed in connection with these fronts. In its present form, the manuscript is extremely vague regarding the method, the presentation of the results, and even the conclusion. More details can be found below. Also some figures should be more focused as well as the captions, otherwise the reader gets lost. I therefore recommend major revision before the paper can be accepted.

- regarding the method : There is no information about the classification method which

is employed : which channels are considered, how, what are the values of the thresholds (if any) ? I would like to see at least one result of the classification (more precise than just figure 2) to try to figure out how it works. It is important since authors mention that they use an improved algorithm. Please specify what is the nature of the improvement and how it has impacted the results of the study. What is the effect of the distorsion for the considered latitudes ? In the same way, authors say that they use TRMM for looking at rainfall (I think that the term trend is misused here). But there are several different TRMM products, so which one has been used ?

- regarding the results : It is not always clear if the points which are discussed are direct results of this study or are general statements. This is especially true for Section 3.1. Therefore I would recommend that the authors reorganize the presentation in order to clarify it. Also Figure 3 represents an extremy vast area, I am not totally convinced that this is useful.

- regarding the last section : I am not sure that the first 4 points are really a result of the study, to me they look more like generalities. The main result is probably point 5, but in its present form, it does not provide a full summary of the results of the paper.

Some other points :

- Page 4, line 17-18 : Authours claim that the Mediterranean frontogenesis shows a direct impact on the European and Asian weather : Please be more specific.

- Page 5, lines 4-6 : is this information necessary ?

- Page 8 : Authors say that the result shows dynamicity of frontogenesis for years 2010 and 2011. Where do I see these results, on which figure ?
* * *

---

## Author Comment (AC1) · 11 Mar 2016

Thank you for your valuable comments. Somehow the reviewer is misinterpret the term "improved scheme" with improved algorithm. Improved scheme relates to the use of large archive of satellite imagery simultaneously in different band wavelengths.

ïČŸ As per understandings of the authors, the main objection raised is of the methodology details. As mentioned in the abstract about the fundamental characteristics identification of fronts and was answered in page 6 Para 2:

"Overlay scheme was used in ArcGIS environment for the identification of exactgeographic locations where these fundamental parameters of frontogenesis meet invisible,

infrared and water vapor bands. The overlay scheme was important as the assertion of frontogenesis should meet the three fundamentals."

In this technique each classified (water vapor, infra-red) and visible band image undergoes visual inspection. For better understanding and results of visual interpretation of frontal process, expert opinion and theoretical literature was used. The visual inspection of images results in the identification of geographical locations of the fundamental characteristics belongs to fronts as mentioned in the manuscript with literature citations.

The identified fundamentals of fronts on visual inspection of each image were overlaid. The geographical location was recorded (for the existence of front) only if all the fundamentals from visible, infra-red and water vapor imagery had same location.

ïČŸ The pattern of the fronts is clearly mentioned under heading 3.2. The selected area is the maximum extent in which the frontal spells in each season were identified by using the visual inspection, classification and overlay schemes. However, methodology as discussed above will be modified in revised manuscript.

ïČŸ While discussing the rainfall trends within geographical delineated extent of frontogenesis emphasis was made on the variation in the trend of peak rainfall. However, to correlate the rainfall with the existence of fronts, a case of Pakistan was putt in concern. The topic of research also explaining the point in itself.

Minor edits highlighted are acknowledged and will be amended in the revised manuscript.

---

## Author Comment (AC2) · 11 Mar 2016

Regarding Methodology ïČŸ Somehow the reviewer has confused the term "improved scheme" with improved algorithm. Improved scheme relates to the use of large archive of satellite imagery simultaneously in different band wavelengths. Whereas, algorithm is far different in this sense and the study didn't use any advance or modify/improve any existing algorithm.

ïČŸ It is clearly mentioned in the manuscript (Page 5 Para 4) that unsupervised classification (K-means) was performed on infrared and water vapor imagery and classes were formed with the variation in brightness values of the image.

[Figure]

ïČŸ The images used are in 3-dimensional domain and as we move towards higher latitudes and east west extremities of the image the scale distorted enormously. However the selected study area has minute variation in the scale that was managed by the manual scale. The manual scale (dividing 10 degree interval into equal and smaller intervals) was incorporated in between each 10 degree interval of the existing scale in the Meteosat-7 satellite imagery.

ïČŸ Rainfall data from the TRMM product 3B42 version 7 with spatial resolution 0.25degree is used. Whereas, the TRMM (3B42 version 7) data was used for 1.5 years and the trend of peak rainfall value was analyzed.

Regarding Results

ïČŸ The result and discussion section is united in a single heading therefore the results are presented simultaneously with discussion.Section 3.1 (line 2-7) presenting the actual results. However, Fig 3 was portray as to show the geographical position of the results in global prospective.

Regarding Last Section:

ïČŸ The conclusion section concluded the research as well as the psyche of frontogenesis not actually the results. However, acknowledged.

Some Other Points:

ïČŸ As to analyze the direct impacts the authors work on the scenario and correlation of fronts and weather event in Pakistan (flood 2010).

ïČŸ The information was mentioned as to describe the extent of study area.

ïČŸ The dynamicity in the extent of frontogenesis is mentioned in 3.2 section, however the related fig 3 presents the results. The legends of the figure shows the seasonal extents of these phenomena.